# Detection and Genotypic Analysis of *Anaplasma bovis* and *A. phagocytophilum* in Horse Blood and Lung Tissue

**DOI:** 10.3390/ijms24043239

**Published:** 2023-02-07

**Authors:** Min-Goo Seo, In-Ohk Ouh, Dongmi Kwak

**Affiliations:** 1College of Veterinary Medicine, Kyungpook National University, 80 Daehak-ro, Buk-gu, Daegu 41566, Republic of Korea; 2National Institute of Health, Korea Disease Control and Prevention Agency, 212 Osongsaengmyeong 2-ro, Cheongju, Chungbuk 28160, Republic of Korea

**Keywords:** *Anaplasma* spp., blood, horse, phylogeny, lung, restriction fragment length polymorphism

## Abstract

A clinical case of *Anaplasma bovis* was reported for the first time in our previous study (2019) in a horse, a nondefinitive host. Although *A. bovis* is a ruminant and not a zoonotic pathogen, it is responsible for persistent infections in horses. In this follow-up study, the prevalence of *Anaplasma* spp., including *A. bovis*, was assessed in horse blood and lung tissue samples to fully understand *Anaplasma* spp. pathogen distribution and the potential risk factors of infection. Among 1696 samples, including 1433 blood samples from farms nationwide and 263 lung tissue samples from horse abattoirs on Jeju Island, a total of 29 samples (1.7%) tested positive for *A. bovis* and 31 (1.8%) samples tested positive for *A. phagocytophilum*, as determined by 16S rRNA nucleotide sequencing and restriction fragment length polymorphism. This study is the first to detect *A. bovis* infection in horse lung tissue samples. Further studies are needed to clarify the comparison of sample types within cohorts. Although the clinical significance of *Anaplasma* infection was not evaluated in this study, our results emphasize the need to clarify the host tropism and genetic divergence of *Anaplasma* to enable the development of effective prevention and control measures through broad epidemiological studies.

## 1. Introduction

*Anaplasma phagocytophilum*, a Gram-negative intracellular bacterium, is the etiologic agent of the tick-borne disease, equine granulocytic anaplasmosis (EGA; formerly known as equine granulocytic ehrlichiosis) [1]. The classic signs of EGA include anorexia, fever, depression, ataxia, petechia, peripheral edema, and reluctance to move [2]. *A. phagocytophilum* belongs to the order Rickettsiales and includes several variants that have distinct host specificities. Despite such preferences, these variants are classified as a single species based on their genetic similarities [1].

To date, multiple *Anaplasma* spp. have been detected in the Republic of Korea using PCR, including *A. phagocytophilum* in dogs [3]; *A. phagocytophilum*, *A. bovis*, and *A. centrale* in ticks [4]; *A. capra*, *A. bovis*, and *Anaplasma* sp. in ticks from water deer [5]; *A. capra*, *A. bovis*, and *A. phagocytophilum*-like *Anaplasma* spp. (APL) clade A and B in ticks from cattle [6]; *A. bovis* [7], *A. phagocytophilum*, APL clade A [8], *A. capra*, and *A. bovis* in cattle [9]; *A. phagocytophilum* and *A. bovis* in wild Korean water deer [10]; *A. phagocytophilum* in a single horse [11]; and a clinical case of *A. bovis* infection in another single horse [12].

Reportedly, 27,829 horses were reared in the Republic of Korea in 2017. Of these, 15,234 (54.7%) were raised on Jeju Island [13] where disease control is of paramount importance for horse farms. Recently, a clinical case of *A. bovis* was reported in a horse, a nondefinitive host, in the Republic of Korea for the first time [12], warranting a follow-up study across the nation, including on Jeju Island. Blood samples are typically used to detect *Anaplasma* spp.; however, in persistently infected animals with intermittent or low-level bacteremia, other tissue samples, such as the liver, lung, lymph nodes, bone marrow, and skin biopsies, are also suitable [14]. *Anaplasma* spp. have been detected in the lung tissues of humans, horses, sheep, cattle, red foxes, and raccoon dogs in several previous studies [15,16,17,18]. Therefore, this follow-up study aimed to detect *Anaplasma* spp. (including *A. bovis*) in both horse blood and lung tissue samples and to elucidate pathogen distribution and potential risk factors in the Republic of Korea.

## 2. Results

### 2.1. Nested PCR and Restriction Fragment Length Polymorphism

In this study, the results of nested PCR (nPCR) amplification of the 16S rRNA gene fragments using the EE1/EE2 and EE3/EE4 primer pairs indicated that 3.5% (60/1696; 53 lung tissue samples and 7 blood samples) of the horses were positive for *Anaplasma* spp. Additional nPCR analyses were conducted to amplify the 16S rRNA fragments of *A. phagocytophilum* and *A. bovis* for species identification. Upon nPCR, horse lung samples that were positive for *A. phagocytophilum* produced fragments that were 641 bp in length (Figure 1A, lane 4), whereas those positive for *A. bovis* generated fragments that were 551 bp in length (Figure 1A, lanes 5 and 6). The samples that were negative for *Anaplasma* spp. (Figure 1A, lanes 2 and 3) did not generate fragments during nPCR. The housekeeping gene (HKG) amplicons from all horse samples generated fragments that were 204 bp in length (Figure 1B, lanes 2 to 10). A sample containing no horse DNA was used as the negative control (Figure 1B, lane 11).

Species-specific PCR indicated that 0.2% (3/1433, 95% confidence interval, CI: 0–0.4) of the blood samples and 9.9% (26/263, 95% CI: 6.3–13.5) of the lung tissue samples were positive for *A. bovis* (Table 1). Moreover, 0.3% (4/1433, 95% CI: 0–0.6) of the blood samples and 10.3% (27/263, 95% CI: 6.6–13.9) of the lung tissue samples were positive for *A. phagocytophilum* (Table 1). When data were analyzed according to the sample groups, a statistically significant difference (*p* = 0.0253) was observed only in geographical regions with *A. phagocytophilum* infections.

The restriction fragment length polymorphism (RFLP) assay was performed by digesting 16S rRNA amplicons (924–926 bp) with *Ale*I for additional discrimination between *A. bovis* and *A. phagocytophilum*. Upon digestion, *A. bovis* amplicons from the horse blood and lung samples generated fragments that were 660 and 264 bp in length, respectively (Figure 2A, lanes 3 and 5), whereas the *A. phagocytophilum* amplicons from the horse lung tissue were not digested by the enzyme (Figure 2A, lane 7). Digestion of the amplicons with *Btg*ZI was also performed to distinguish *A. bovis* from *A. phagocytophilum*. The *A. bovis* amplicons from the horse blood and lung tissue samples could not be digested by the enzyme (Figure 2B, lanes 3 and 5), whereas the *A. phagocytophilum* amplicon from the horse lung tissue samples generated two fragments of 707 and 223 bp (Figure 2B, lane 7). Interestingly, no samples were co-infected with these or any other *Anaplasma* species.

### 2.2. Cloning, Sequencing, and Phylogeny

The RFLP assay data were confirmed by the sequencing of the amplicons from the species-specific nPCR of 16S rRNA. All positive samples were confirmed via RFLP. As the *A. bovis* sequences shared 99.7–100% identity (differences in 0–3 nucleotide positions), representative lung tissue sequences were randomly selected for phylogenetic analysis. The included samples were as follows: blood (H-B-JJ-32, H-B-JJ-179, and H-B-GG-300; GenBank accession numbers MK028574, MT800792, and MT800793, respectively) and lung (H-L-JJ-46, H-L-JJ-47, and H-L-JJ-48; accession numbers MK028571, MK028572, and MK028573, respectively). Phylogenetic analysis revealed that the *A. bovis* sequences (Figure 3) were clustered with previously documented sequences.

Nucleotide sequencing of the amplicons revealed that four blood and 27 lung tissue samples were positive for *A. phagocytophilum*. As the *A. phagocytophilum* 16S rRNA sequences shared 99.4–100% identity (differences in 0–5 nucleotide positions), representative lung tissue sequences were randomly selected for the phylogenetic analyses. The analyzed samples were as follows: blood (H-B-JJ-306, H-B-GW-370, H-B-WS-392, and H-B-BS-399; accession numbers MN650821, MT800789, MT800790, and MT800791, respectively) and lung (H-L-JJ-18, H-L-JJ-27, H-L-JJ-28, and H-L-JJ-31; accession numbers MK811372, MK811373, MK811374, and MK811375, respectively). The phylogenetic analysis revealed that the sequences of *A. phagocytophilum* (Figure 3) clustered with previously documented sequences.

Six *A. bovis* samples (H-B-JJ-32, H-B-JJ-179, H-B-GG-300, H-L-JJ-46, H-L-JJ-47, and H-L-JJ-48) belonging to clade B shared 94.1–100% sequence identity with previously identified isolates collected from cattle in Tunisia (94.1%, KM401902), a horse in Japan (95.2%, JX082006), a sika deer in China (99.8%, KJ659040), a tick in the Republic of Korea (99.1%, AF470698), and a horse in the Republic of Korea (100%, MH794247). Samples clustered in clade A shared 98.1–99.9% sequence identity; these samples included isolates from cattle in the Republic of Korea (98.1%, MF197897), a tick in China (99.5%, KP314250), and a tick in the Republic of Korea (99.9%, GU064901).

Eight *A. phagocytophilum* samples (H-B-JJ-306, H-B-GW-370, H-B-WS-392, H-B-BS-399, H-L-JJ-18, H-L-JJ-27, H-L-JJ-28, and H-L-JJ-31) shared 98.9–99.9% sequence identity with previously collected *A. phagocytophilum* isolates from a horse in the Netherlands (98.9%, KF242656), a horse in the United States (99.2%, AF172165), a horse in China (99.6%, GQ900617), and a horse in the Republic of Korea (99.9%, MH794246).

## 3. Discussion

A previous molecular study identified *A. phagocytophilum* infection in a horse in the Republic of Korea [11]. A clinical case of *A. bovis* infection in a horse, a nondefinitive host, was also identified [12]. In this study, we screened horses for the presence of *A. bovis*; 3 (0.2%) blood and 26 (9.9%) lung tissue samples from 1696 animals were found to be positive based on a species-specific nPCR analysis of 16S rRNA gene fragments. *A. phagocytophilum* was also detected in horse blood (*n* = 4; 0.3%) and lung tissue (*n* = 27; 10.3%) samples. These positive samples were further confirmed to be *A. bovis* and *A. phagocytophilum* via the RFLP assay.

Cross-contamination is a major problem associated with nPCR. Reducing the potential for contamination and ensuring the differentiation between true- and false-positive results necessitates the incorporation of reference genes as internal controls [20]. Reference genes, commonly known as HKG, are stably expressed in cells and tissues. An internal reference gene helps to ensure the accurate interpretation of the data [21]. Although several HKGs have been used, the 18S rRNA gene is considered as one of the most stable genes [19] that shows limited changes in expression under different experimental conditions [22]. Moreover, 18S rRNA sequences are highly conserved among eukaryotes, and a single assay can be used for an HKG measurement even in studies involving cells from several species [22]. In this study, horse 18S rRNA was used as the internal positive control and *Coxiella burnetii* was used as the internal negative control to ensure that cross-contamination did not occur and to improve the accuracy of the data.

Equine anaplasmosis is typically caused by *A. phagocytophilum*, the causative zoonotic pathogen of granulocytic anaplasmosis and tick-borne fever [23]. However, in this study, we detected *A. bovis* in the lung tissues of horses for the first time. *A. bovis* is widespread in tropical and subtropical regions [24] and is a common ruminant pathogen infecting buffalo and cattle in Africa and Asia [25]. Infections in other hosts have been rarely detected, such as in a leopard in the Republic of Korea [26], a Hokkaido brown bear in Japan [27], a sika deer and a wild boar in Japan [28], a raccoon in Japan [29]; moreover, human cases have been reported in China [30].

Unfortunately, the lung tissue samples collected from abattoirs were not associated with any demographic information. Nonetheless, as the blood and lung tissue samples obtained from Jeju Island tested positive for *A. bovis* and *A. phagocytophilum*, we surmised that *Anaplasma* spp. may be widespread on Jeju Island. This hypothesis is supported by multiple reports of the molecular detection of *Anaplasma* spp. in animals on Jeju Island. The prevalence has been reported for *A. phagocytophilum* (1.9%, 27/1395), *A. bovis* (0.4%, 5/1395), and *A. centrale* (0.1%, 1/1395) in the tick *Haemaphysalis longicornis* [4]; *A. bovis* (13.9%, 10/72) in *H. longicornis* from grazing cattle [31]; *Anaplasma* spp. (18%, 7/39) in native Korean goats [32]; and *A. bovis* (4.2%, 3/71) in cattle [7].

The prevalence of *Anaplasma* spp. has been reported to significantly differ between geographic locations and is associated with tick habitats and distribution [33,34]. Prevalence has also been reported to vary according to biogeoclimatic conditions. Cattle from sub-humid areas have been reported to be more susceptible to *Anaplasma* infection compared to cattle from semi-arid regions [34]. The climate of the Korean peninsula is steadily turning subtropical; Jeju Island is particularly vulnerable to this change because of its lower latitude. *Haemaphysalis longicornis* is the most commonly identified tick species in Korea and is more abundantly distributed across Jeju Island than the mainland [4]. The horses in this environment are allowed to graze freely in grassland areas, thereby increasing their exposure to ticks compared with those living on the mainland, which are kept in restricted areas. Biogeoclimatic differences may thus affect the prevalence of ticks as well as tick-borne infections. On the mainland, the southern region of the Republic of Korea exhibits by far the highest prevalence of *A. phagocytophilum*. Moreover, the horse industry continues to grow in the Republic of Korea each year. Therefore, it is extremely important to mitigate or prevent the spread of zoonotic diseases, such as EGA, between horses and humans on Jeju Island.

Blood smears from animals experiencing persistent infections may yield negative infection results, even though 400 granulocytes are adequate to detect infected leukocytes in ruminants with a history of recent illness [35,36]. Moreover, a negative result does not preclude infection. Microscopy in addition to supplementary diagnostic laboratory methods and the screening of other sample types are recommended if persistent infection is suspected [14]. In postmortem evaluations, tissue impressions or smears from the liver, spleen, kidneys, lungs, heart, or blood vessels can be used to visualize erythrocytotropic *Anaplasma* spp. [37]. In this study, blood sample analyses demonstrated a lower prevalence of both *A. bovis* (0.2%) and *A. phagocytophilum* (0.3%), whereas lung tissue samples had a higher prevalence of both *A. bovis* (9.9%) and *A. phagocytophilum* (10.3%). Unfortunately, the collection of blood and lung samples from different animals precludes the comparison of the prevalence of these pathogens between sample types. Further studies are needed to compare the sample types within cohorts. At the time of our investigation, the lung tissue samples were rich in blood, which might have resulted in the improved detection rates of *A. bovis* and *A. phagocytophilum*. These results suggest that *A. bovis* and *A. phagocytophilum* cause frequent and persistent infections in horses.

Although *Anaplasma* spp. usually infects blood cells, their presence in several other tissues has been reported. *A. phagocytophilum* has been detected in the kidney, thymus, sternal bone marrow, small intestine, mediastinal lymph node, and bladder wall tissue of persistently infected sheep [36]. In another study on humans, horses, and sheep, *A. phagocytophilum* was detected in the lungs, spleen, and liver; moreover, large numbers of infected neutrophils were detected in blood vessel lumens, mainly in the microvasculature of the lungs or in the sinusoids of the red pulp of the spleen [15]. *A. phagocytophilum* was also detected in the lung tissues of red foxes and raccoon dogs [18] and in the lung, spleen, liver, and heart tissue of cattle [16]. In another case, *A. marginale* was detected in the lymph node, spleen, heart, lung, and ear skin of cattle [16]. In humans, anaplasmosis caused by *A. phagocytophilum* presents as atypical pneumonitis and histopathological changes in the lungs [38]. Consistent with these reports, *A. bovis* and *A. phagocytophilum* were both detected in horse lung tissue samples in the current study. *A. phagocytophilum* infection in tissue has been associated with infected circulatory neutrophils rather than infected tissue cells [36]. Similarly, *A. bovis* detected in the current study might have infected circulatory monocytes within the lung tissue.

Based on phylogenetic analyses using 16S rRNA sequences, *A. bovis* was classified into clades A and B. The *A. bovis* sequences identified in this study belonged to clade B, which includes the strains identified in Tunisia, the Republic of Korea, the Democratic People’s Republic of Korea, China, and Japan. The *A. bovis* sequences of clade A contain strains from East Asia (China and the Republic of Korea). These results are consistent with an earlier study in which a genotypic analysis of *Anaplasma* spp. revealed a high degree of identity with species isolated from neighboring countries [39]. The sequences of 16S rRNA genes from *Anaplama* spp. have been reported for several species on Jeju Island: *A. bovis* in *H. longicornis* (EU181143, clade B; GU064901, clade A), *A. bovis* in cattle (MF197897, clade A), and *A. phagocytophilum* in *H. longicornis* from a horse (AF470700). The horse-derived *A. bovis* sequences in this study showed high sequence identities with those previously reported on Jeju Island.

To the best of our knowledge, this is the first study to perform a molecular detection of *A. bovis* in horse lung tissues. Recently, infected nondefinitive hosts were reported as threats for the spread of several diseases to humans. Indeed, human cases of goat-derived *A. capra* in China [40], sheep-derived *A. ovis* in one patient in Cyprus [41], and cattle-derived *A. bovis* in two patients in China [30] have been reported previously. *A. bovis* is a ruminant pathogen and, unlike *A. phagocytophilum*, it is not considered a zoonotic pathogen. However, *A. bovis* may cause frequent and persistent infections in horses (a nondefinitive host) reared on Jeju Island. Our study can be used as a reference for further investigating the clinical significance of *Anaplasma* infections. This study highlights the need for broad epidemiological studies to clarify the host–pathogen relationship and genetic divergence of *Anaplasma* spp. to aid in the development of effective preventive and control measures.

## 4. Materials and Methods

### 4.1. Ethical Approval

This study was conducted between 2017 and 2019 and did not need the approval of the Institutional Animal Care and Use Committee at Kyungpook National University as this is only required for research involving laboratory animals kept in indoor facilities, not outdoor animals. Practicing veterinarians collected whole blood samples during treatment or at regular medical check-ups, after obtaining verbal consent from the farmers.

### 4.2. Sample Size Determination and Sample Collection

Using a simple random sampling strategy and considering an expected disease prevalence of 10%, accepted absolute error of 5%, and confidence level of 95%, the sample size was calculated using the following formula [42]:$$n=\frac{1.96^{2}p_{exp}\left(1-p_{exp}\right)}{d^{2}}$$
where *n* represents the required sample size, *d* represents the desired absolute precision, and *p_exp_* represents the expected prevalence.

The formula indicated that the statistical power required a minimum of 138 samples. In this study, we collected 1696 samples (1433 blood and 263 lung tissue samples) from horses across the country, including Jeju Island, between 2017 and 2019 (Figure 4). Whole blood samples were collected from horse farms nationwide, whereas lung tissue samples were randomly collected from horse abattoirs on Jeju Island. Horse meat is a popular delicacy on Jeju Island and raw horse meat is regularly consumed here. Data on age, sex, location, activity, and breed were recorded for each blood sample; no additional data were recorded for the lung tissue samples.

### 4.3. DNA Extraction and PCR

Genomic DNA was extracted from the whole blood and lung tissue samples using a DNeasy Blood & Tissue Kit (Qiagen, Melbourne, Australia) following the manufacturer’s instructions. The extracted DNA was kept at −20 °C before use. PCR amplification was carried out using an AccuPower HotStart PCR Premix Kit (Bioneer, Daejeon, Republic of Korea). First, infection with *Anaplasma* spp. was screened via the amplification of 16S rRNA fragments using nPCR with the primer pairs EE1/EE2 and EE3/EE4 to obtain an expected amplicon of 924–926 bp in length [8,9]. For species identification, the 16S rRNA genes of *A. phagocytophilum* and *A. bovis* were identified by re-amplifying the PCR-positive samples using the primer pairs EE1/EE2 and SSAP2f/SSAP2r and EE1/EE2 and AB1f/AB1r, respectively [9], to obtain the expected amplicons of 641 and 551 bp in length, respectively. The positive control for each PCR reaction comprised the *A. phagocytophilum* [11] and *A. bovis* [12] strains that were previously identified in horses from mainland Korea. For each PCR reaction, a sample with distilled water and PCR reagents but no DNA was used as the negative control.

We addressed concerns regarding the amplicon contamination of nPCR using a distinct positive control sequence to ensure the differentiation of true positive results from those caused by possible contamination. The HKG of the 18S rRNA expressed with high stability in horse tissue and cultured cells [19] was used as an internal positive control. To identify the 18S rRNA HKG, horse DNA samples were amplified using previously published primer sequences to obtain the expected 204 bp long amplicons [19]. A horse blood sample infected with *C. burnetii* [43] was used as the internal negative control using the primer sets EE1/EE2 and SSAP2f/SSAP2r and EE1/EE2 and AB1f/AB1r.

All primers and amplification conditions used in the present study are presented in Appendix A.

### 4.4. RFLP

*A. phagocytophilum* and *A. bovis* were identified by digesting the 16S rRNA nPCR products of 868–870 bp in length (the PCR amplicon of 924–926 bp without primer sequences) using two restriction enzymes for the RFLP assay [9]. The restriction enzymes *Ale*I and *Btg*ZI were used for the RFLP assay conducted using the CLC Main Workbench 6.7.2 (CLC Bio, Qiagen, Aarhus, Denmark). The solution subjected restriction digestion comprised 10 μL of PCR product, 5 μL of buffer (10×, 1 μL of *Ale*I (10,000 U/mL; New England Biolabs, Hitchin, UK) or 2 μL of *Btg*ZI (5000 U/mL; New England Biolabs), and distilled water to obtain a final volume of 50 μL. For *Btg*ZI or *Ale*I, the reactions were incubated for 1 h at 60 °C or 30 °C, respectively. The restricted fragments were separated through electrophoresis on a 3% agarose gel in TAE solution at 100 V for 60 min. The gel was then stained with ethidium bromide and subjected to UV visualization.

### 4.5. DNA Cloning

The QIAquick Gel Extraction Kit (Qiagen) was used to purify the PCR products from the positive reactions produced using the 16S rRNA primers EE3/EE4. Using the pGEM-T Easy vector (Promega, Madison, WI, USA), the purified products were ligated as per the manufacturer’s recommendations. Competent *Escherichia coli* DH5α cells were transformed from the ligation product and were incubated at 37 °C overnight. As directed by the manufacturer, plasmid DNA was extracted using a plasmid miniprep kit (Qiagen).

### 4.6. DNA Sequencing and Phylogenetic Analysis

A few recombinant clones were chosen and delivered to Macrogen (Seoul, Republic of Korea) for sequencing. CLUSTAL Omega (v. 1.2.1, http://www.clustal.org/omega/, accessed on 1 September 2022) was used to align and evaluate the sequences; then, the alignment was modified using BioEdit (v. 7.2.5, http://www.mbio.ncsu.edu/BioEdit/bioedit.html, accessed on 1 September 2022). Phylogenetic analysis was conducted using MEGA (v. 6.0, https://megasoftware.net, accessed on 1 September 2022) through the maximum likelihood technique. The aligned sequences were examined by developing a similarity matrix, and a bootstrap approach with 1,000 repeats was used to determine the stability of the trees.

### 4.7. Statistical Analysis

Pearson’s chi-squared test was used to examine the significant differences between the groups. A *p*-value of <0.05 was considered as statistically significant. All statistical calculations were performed using the statistical analysis program GraphPad Prism (v. 5.04; GraphPad Soft-ware Inc., La Jolla, CA, USA). All estimations were given a 95% CI.

## Figures and Tables

**Figure 1 ijms-24-03239-f001:**
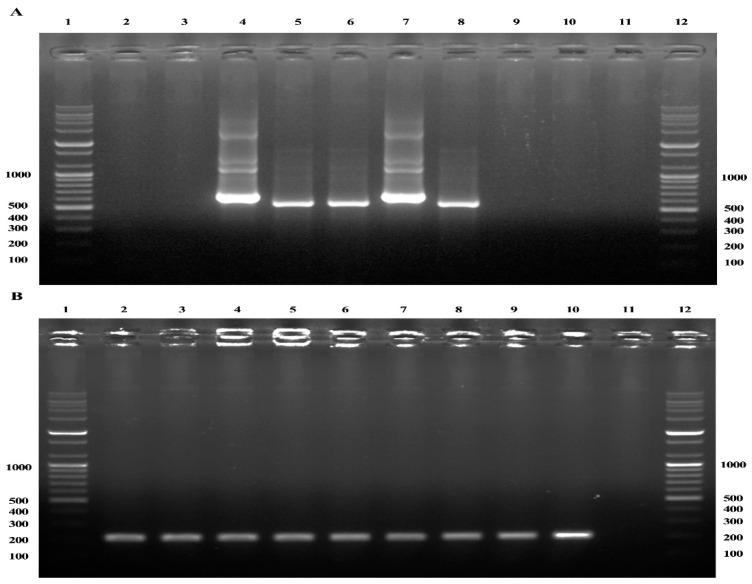
PCR detection of *Anaplasma* 16S rRNA and horse housekeeping gene (HKG) 18S rRNA in horse blood and lung tissue samples. (**A**) Nested PCR was used to identify the 16S rRNA genes of *A. phagocytophilum* and *A. bovis* using the primer sets EE1/EE2 and SSAP2f/SSAP2r and EE1/EE2 and AB1f/AB1r, respectively. Lanes 1 and 12: 100 bp ladder; lanes 2 and 3: horse blood and lung tissue samples negative for *Anaplasma* spp. using the primers EE1/EE2 and SSAP2f/SSAP2r or AB1f/AB1r, respectively; lane 4: *A. phagocytophilum* PCR product (641 bp) from horse lung sample; lane 5: *A. bovis* PCR product (551 bp) from horse blood samples; lane 6: *A. bovis* PCR product (551 bp) from horse lung tissue samples; lane 7: PCR product (641 bp) of *A. phagocytophilum* previously detected in a horse (positive control); lane 8: PCR product (551 bp) of *A. bovis* previously detected in a horse (positive control); lanes 9 and 10: internal negative control samples of *Coxiella burnetii* previously detected in a horse using the primer sets EE1/EE2 and SSAP2f/SSAP2r and EE1/EE2 and AB1f/AB1r, respectively; lane 11: lack of an amplicon from horse DNA generated using primers EE1/EE2 and SSAP2f/SSAP2r/AB1f/AB1r. (**B**) Single-round PCR detection using primers specific to horse HKG 18S rRNA. Lanes are in the same order as described in (**A**) but previously published primer sequences were used, producing expected amplicons of 204 bp [19].

**Figure 2 ijms-24-03239-f002:**
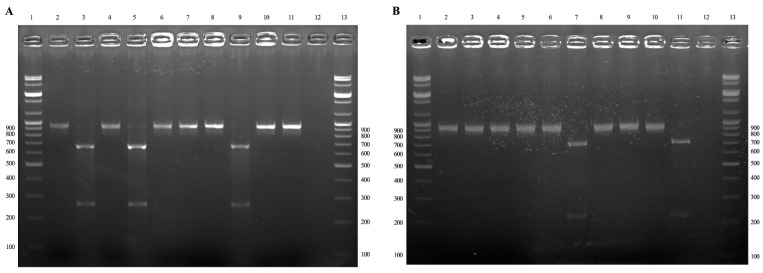
Restriction fragment length polymorphism (RFLP) assay and *Anaplasma* species identification. (**A**) RFLP analysis using *Ale*I enzyme for PCR products. Lanes 1 and 13: 100 bp ladder; lanes 2 and 3: *A. bovis* PCR products from horse blood samples before (924 bp) and after (660 and 264 bp) *Ale*I restriction digestion; lanes 4 and 5: *A. bovis* PCR products from horse lung tissue samples before (924 bp) and after (660 and 264 bp) *Ale*I restriction digestion; lanes 6 and 7: *A. phagocytophilum* PCR products from horse lung tissue samples before (926 bp) and after (926 bp) *Ale*I restriction digestion; lanes 8 and 9: PCR products of *A. bovis* detected in horse (positive control) before (924 bp) and after (660 and 264 bp) *Ale*I restriction digestion; lanes 10 and 11: PCR products of *A. phagocytophilum* detected in horse (positive control) before (926 bp) and after (926 bp) *Ale*I restriction digestion; lane 12: negative control. (**B**) RFLP analysis using the *Btg*ZI enzyme for PCR products. Lanes 1 and 13: 100 bp ladder; lanes 2 and 3: *A. bovis* PCR products from horse blood samples before (924 bp) and after (924 bp) *Btg*ZI restriction digestion; lanes 4 and 5: *A. bovis* PCR products from horse lung tissue samples before (924 bp) and after (924 bp) *Btg*ZI restriction digestion; lanes 6 and 7: *A. phagocytophilum* PCR products from horse lung tissue samples before (926 bp) and after (707 and 223 bp) *Btg*ZI restriction digestion; lanes 8 and 9: PCR products of *A. bovis* detected in horse (positive control) before (924 bp) and after (924 bp) *Btg*ZI restriction digestion; lanes 10 and 11: PCR products of *A. phagocytophilum* detected in horse (positive control) before (926 bp) and after (707 and 223 bp) *Btg*ZI restriction digestion; lane 12: negative control.

**Figure 3 ijms-24-03239-f003:**
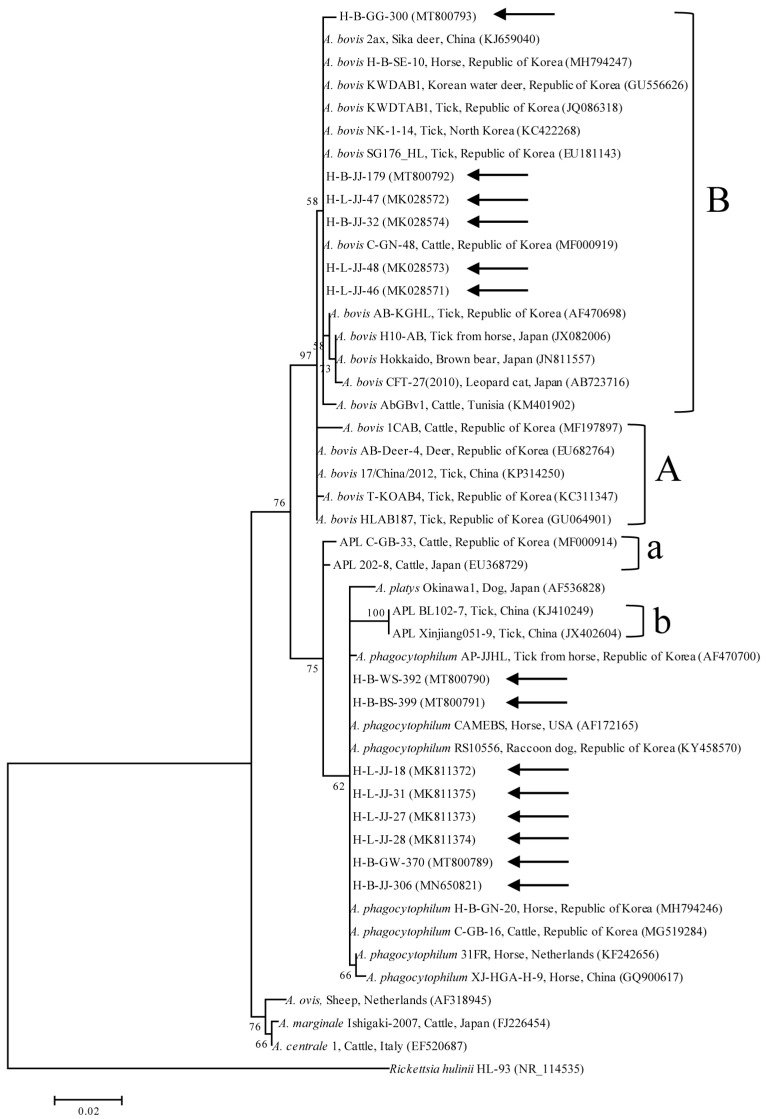
Phylogenetic tree created using the maximum likelihood technique with 16S rRNA nucleotide sequences from *Anaplasma*. The sequences analyzed in this study are marked with arrows. The GenBank accession numbers of the additional sequences are listed adjacent to their respective sequence names. Branch numbers indicate bootstrap support (1000 replicates). The scale bar represents the phylogenetic distance between the sequences. *A. bovis* clusters are denoted using “A” and “B.” *A. phagocytophilum*-like *Anaplasma* spp. clusters are denoted using “a” and “b.”

**Figure 4 ijms-24-03239-f004:**
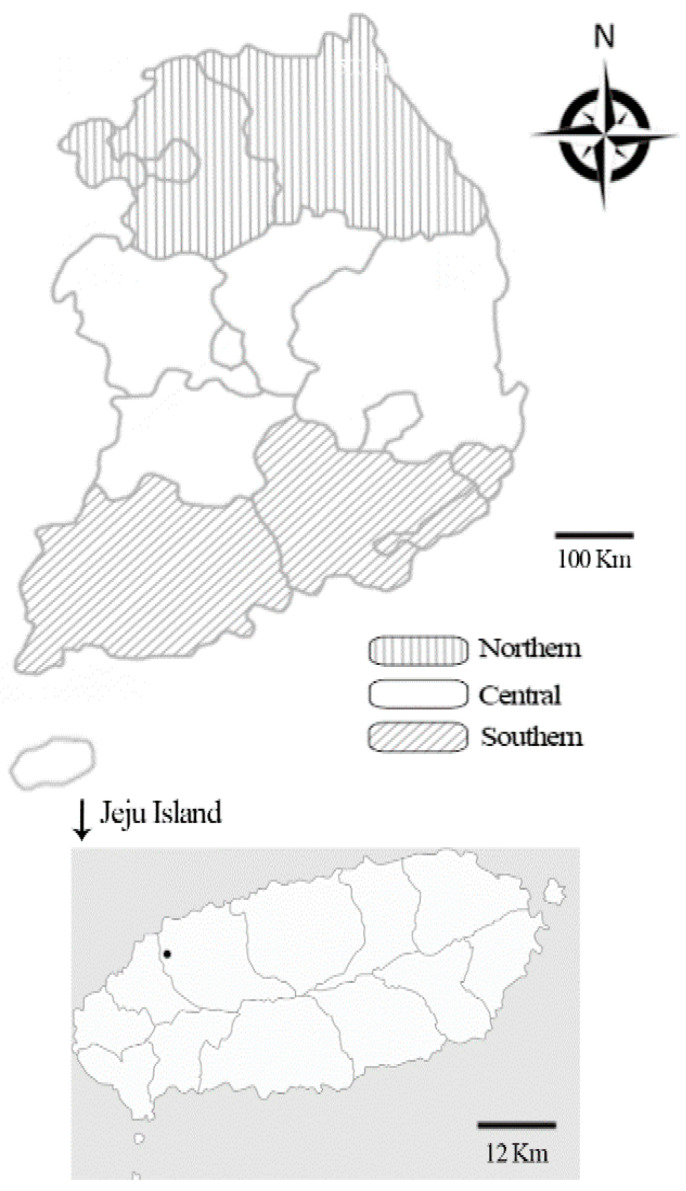
Map of the Republic of Korea. Horse blood samples were collected throughout the nation; however, lung tissue samples were only collected from Jeju Island. The dot denotes the location of the abattoir on Jeju Island from which the horse lung samples were collected. The regions of sample collection were divided into four groups: northern, central, southern, and Jeju Island.

**Table 1 ijms-24-03239-t001:** Prevalence of *Anaplasma bovis* and *A. phagocytophilum* as determined by the analysis of 16S rRNA genes in horse samples from the Republic of Korea between 2017 and 2019.

Sample		Group	No. Tested	*A. bovis*			*A. phagocytophilum*		
				No. Positive (%)	95% CI ^†^	*p*-Value	No. Positive (%)	95% CI ^†^	*p*-Value
Blood	Sex	Female	756	2 (0.3)	0–0.6	0.6040	3 (0.4)	0–0.8	0.5021
		Male	354	0	0		0	0	
		Castrated	323	1 (0.3)	0–0.9		1 (0.3)	0–0.9	
	Age (year)	<5	460	0	0	0.3204	0	0	0.3752
		5–10	537	1 (0.2)	0–0.6		2 (0.4)	0–0.9	
		>10	436	2 (0.5)	0–1.1		2 (0.5)	0–1.1	
	Breed	Thoroughbred	456	2 (0.4)	0–1.0	0.4324	3 (0.7)	0–1.4	0.2361
		Warmblood	308	1 (0.3)	0–1.0		1 (0.3)	0–1.0	
		Native Korean Pony	416	0	0		0	0	
		Mixed	253	0	0		0	0	
	Type of activity	Broodmare	549	2 (0.4)	0–0.9	0.6730	1 (0.2)	0–0.5	0.2843
		Leisure	452	1 (0.2)	0–0.7		3 (0.7)	0–1.4	
		Race	210	0	0		0	0	
		Breeding	222	0	0		0	0	
	Region	Northern	301	1 (0.3)	0–1.0	0.8016	1 (0.3)	0–1.0	0.0253 *
		Central	216	0	0		0	0	
		Southern	123	0	0		2 (1.6)	0–3.9	
		Jeju Island	793	2 (0.3)	0–0.6		1 (0.1)	0–0.4	
	Subtotal		1433	3 (0.2)	0–0.4		4 (0.3)	0–0.6	
Lung			263	26 (9.9)	6.3–13.5		27 (10.3)	6.6–13.9	
Total			1696	29 (1.7)	1.1–2.3		31 (1.8)	1.2–2.5	

^†^ CI: Confidence interval. * Statistical significance was observed.

## Data Availability

Data supporting the conclusions of this article are included within the article. The newly generated sequences were submitted to the GenBank database under the accession numbers MT800789-93, MK028571-74, MN650821, and MK811372-75. The datasets used and/or analyzed during the present study are available from the corresponding author upon reasonable request.

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
