# Peer review of "Detection and Genotypic Analysis of *Anaplasma bovis* and *A. phagocytophilum* in Horse Blood and Lung Tissue"

_ijms, 2023, doi:10.3390/ijms24043239_

Round 1
Reviewer 1 Report
Manuscript Review
Detection and genotypic analysis of Anaplasma bovis and A. 2 phagocytophilum in horse blood and lung tissue
A clinical case of Anaplasma bovis was reported for the first time in the current study. And the genotypic was assessed in horse blood and lung tissues. It can provide a basis for the development of effective prevention and measures.
Below are my suggestions.
1. Materials and Methods
4.3 DNA extraction and PCR
Please list all primers used in this study.
4.7 Statistical analysis
What statistical software is used for the Pearson’s chi-squared test?
2. Results
Figure1: In lane 4 and 7, What are the other two bands besides the 641bp fragment?
Line 142 and Line 145-146: "are" should be replaced with "were".
Line 143: "indicate" should be replaced with "indicated".
Line 144: "represents" should be replaced with "represented".

Author Response
-Reviewer 1
Detection and genotypic analysis of Anaplasma bovis and A. phagocytophilum in horse blood and lung tissue
A clinical case of Anaplasma bovis was reported for the first time in the current study. And the genotypic was assessed in horse blood and lung tissues. It can provide a basis for the development of effective prevention and measures.
Below are my suggestions.
- Materials and Methods
4.3 DNA extraction and PCR
Please list all primers used in this study.
Response: We appreciate your relevant comment. According to your suggestions, we added all primer in Table S1 and added the sentences as “All primers and amplification conditions used in the present study are presented in Supplementary Table S1.” at lines 340-341 and “Supplementary Materials: The following are available online at www.mdpi.com/xxx/s1, Table S1: Primers used for the detection of Anaplasma spp. and housekeeping gene from horses in the present study.” at lines 375-377.
4.7 Statistical analysis
What statistical software is used for the Pearson’s chi-squared test?
Response: We appreciate your relevant comment. We used statistical analysis software “GraphPad Prism”. We already described in manuscript and changed the sentence more clearly as “All statistical calculations were performed using the statistical analysis program GraphPad Prism (v. 5.04; GraphPad Soft-ware Inc., La Jolla, CA, USA).” at lines 371-373.
- Results
Figure1: In lane 4 and 7, What are the other two bands besides the 641bp fragment?
Response: We appreciate your relevant comment. Lanes 5 and 6 are A. bovis PCR products (551 bp) from horse blood and lung samples, respectively. We already described in manuscript and changed the sentence more clearly as “lane 5: A. bovis PCR product (551 bp) from horse blood samples; lane 6: A. bovis PCR product (551 bp) from horse lung samples” at lines 76-78.
Line 142 and Line 145-146: "are" should be replaced with "were".
Response: Yes, we did as suggested in L146 and L149-150.
Line 143: "indicate" should be replaced with "indicated".
Response: Yes, we did as suggested in L147.
Line 144: "represents" should be replaced with "represented".
Response: Yes, we did as suggested in L148.

Reviewer 2 Report
This research article provides a molecular epidemiological study of Anaplasma phagocytophilum and A. bovis in horses from South Korea. The study design is robust, and the manuscript is well-structured and easy to read. There are some issues that should be improved.
Improvement suggestions:
- Abstract, line 11: please, remove “for the first time”, because there is a previous study that reported a clinical case of A. bovis in a horse, as you mentioned later (Set et al., 2019). Or specify that it is the first time that A. bovis is detected in lung samples from horses.
- Abstract, line 20: please, change “warranted” to “needed”.
- Introduction, line 38: please, define here “Anaplasma sp., (APL)” and remove it from line 39.
- Results, line 57: it would be interesting to add the number of positive blood and lung samples, separately.
- Results, lines 62-63: the sentence “Coxiella burnetii was used as the internal negative control (Fig 1A, lanes 9 and 10)” could be removed, as this information is described in Figure 1 (line 78).
- Discussion, lines 204-207: please, use the same order in all parenthesis. For example, “(percentage, proportion)”.
- Discussion, lines 212-221: Southern region of South Korea should be highlighted at this point, as this area has the significantly highest prevalence of A. phagocytophilum.
- Discussion, line 234: please, change “warranted” to “needed”.
- Discussion, lines 238-260: this paragraph is too long and sometimes difficult to read. Please, try to reduce it.
- Material and methods, line 293: why did you use an expected disease prevalence of 10%? Are there previous epidemiological studies in horses from the same geographical area? If so, please, add references. If not, you should use an unknown expected prevalence (worst scenario, 50%). In this case, sample size using lung samples from Jeju Island may not be enough for estimating an accurate prevalence, although you can provide the ratio of positive animals, and suggest that an epidemiological study with a larger sample size is necessary in this area.
- Material and methods, lines 324-325: which kind of “sample without DNA” was used as the negative control? Please, further description would be necessary.
Author Response
Comments from the reviewer:
-Reviewer 2
This research article provides a molecular epidemiological study of Anaplasma phagocytophilum and A. bovis in horses from South Korea. The study design is robust, and the manuscript is well-structured and easy to read. There are some issues that should be improved.
Improvement suggestions:
- Abstract, line 11: please, remove “for the first time”, because there is a previous study that reported a clinical case of A. bovis in a horse, as you mentioned later (Set et al., 2019). Or specify that it is the first time that A. bovis is detected in lung samples from horses.
Response: We appreciate your critical comment. We want to describe that A. bovis was first detected in horse in our previous study. According to your suggestion, we changed the sentence more clearly as “A clinical case of Anaplasma bovis was reported for the first time in our previous study (2019) in a horse, a non-definitive host.” at lines 11-12.
- Abstract, line 20: please, change “warranted” to “needed”.
Response: Yes, we did as suggested in L20.
- Introduction, line 38: please, define here “Anaplasma sp., (APL)” and remove it from line 39.
Response: Yes, we did as suggested in L39-40.
- Results, line 57: it would be interesting to add the number of positive blood and lung samples, separately.
Response: We appreciate your critical comment. According to your suggestion, we added the sentence more clearly as “In this study, the results of nested PCR (nPCR) amplification of the 16S rRNA gene fragments using the EE1/EE2 and EE3/EE4 primer pairs indicated that 3.5% (60/1,696; 53 lung tissue samples and 7 blood samples) of the horses were positive for Anaplasma spp.” at lines 57-59.
- Results, lines 62-63: the sentence “Coxiella burnetii was used as the internal negative control (Fig 1A, lanes 9 and 10)” could be removed, as this information is described in Figure 1 (line 78).
Response: Yes, we deleted sentence in manuscript as suggested.
- Discussion, lines 204-207: please, use the same order in all parenthesis. For example, “(percentage, proportion)”.
Response: Yes, we did as suggested in L210-213.
- Discussion, lines 212-221: Southern region of South Korea should be highlighted at this point, as this area has the significantly highest prevalence of A. phagocytophilum.
Response: We appreciate your critical comment. According to your suggestion, we added the sentences as “On the mainland, the southern region of South Korea exhibits the significantly highest prevalence of A. phagocytophilum.” at lines 225-226.
- Discussion, line 234: please, change “warranted” to “needed”.
Response: Yes, we did as suggested in L242.
- Discussion, lines 238-260: this paragraph is too long and sometimes difficult to read. Please, try to reduce it.
Response: We appreciate your critical comment. According to your suggestion, we changed the sentence more clearly as “Although Anaplasma spp. usually infects blood cells, their presence in several other tissues has been reported. A. phagocytophilum has been detected in the kidney, thymus, sternal bone marrow, small intestine, mediastinal lymph node, and bladder wall tissue of persistently infected sheep [36]. In another study on humans, horses, and sheep, A. phagocytophilum was detected in the lungs, spleen, and liver; moreover, large numbers of infected neutrophils were detected in blood vessel lumens, mainly in the microvasculature of the lungs or in the sinusoids of the red pulp of the spleen [15]. A. phagocytophilum was also detected in the lung tissues of red foxes and raccoon dogs [18] and in the lung, spleen, liver, and heart tissue of cattle [16]. In another case, A. marginale was detected in the lymph node, spleen, heart, lung, and ear skin of cattle [16]. In humans, anaplasmosis caused by A. phagocytophilum presents as atypical pneumonitis and histopathological changes in the lungs [38]. Consistent with these reports, A. bovis and A. phagocytophilum were both detected in horse lung tissue samples in the current study. A. phagocytophilum infection in tissue has been associated with infected circulatory neutrophils rather than infected tissue cells [36]. Similarly, A. bovis detected in the current study might have infected circulatory monocytes within the lung tissue.” at lines 247-262.
- Material and methods, line 293: why did you use an expected disease prevalence of 10%? Are there previous epidemiological studies in horses from the same geographical area? If so, please, add references. If not, you should use an unknown expected prevalence (worst scenario, 50%). In this case, sample size using lung samples from Jeju Island may not be enough for estimating an accurate prevalence, although you can provide the ratio of positive animals, and suggest that an epidemiological study with a larger sample size is necessary in this area.
Response: We appreciate your critical comment. In our previous study during 2016-2017, one horse was detected A. phagocytophilum (0.2%, 1/627) in blood samples from the mainland of South Korea (Seo et al., Korean J Parasitol 2018). And, only one horse was presented clinical signs of A. bovis in 2017 in South Korea (Seo et al., Vet Microbiol 2019). So, following present study during 2017-2019, we used lowest expected disease prevalence of 10% as mentioned in the reference “Thrusfield, M.V.; Christley, R. Veterinary epidemiology, Fourth edition. ed.; Wiley: Hoboken, NJ, 2018”. I attached related figure (Table 13.2) in the below.
- Material and methods, lines 324-325: which kind of “sample without DNA” was used as the negative control? Please, further description would be necessary.
Response: We appreciate your critical comment. According to your suggestion, we changed the sentence more clearly as “For each PCR reaction, a sample with distilled water and PCR reagents but no DNA was used as the negative control.” at lines 330-331.
